# INFERRING PRINCIPAL COMPONENTS IN THE SIMPLEX WITH MULTINOMIAL VARIATIONAL AUTOENCODERS

## 1 ABSTRACT

Covariance estimation on high-dimensional data is a central challenge across multiple scientific disciplines. Sparse high-dimensional count data, frequently encountered in biological applications such as DNA sequencing and proteomics, are often well modeled using multinomial logistic normal models. In many cases, these datasets are also compositional, presented item-wise as fractions of a normalized total, due to measurement and instrument constraints. In compositional settings, three key factors limit the ability of these models to estimate covariance: (1) the computational complexity of inverting high-dimensional covariance matrices, (2) the non-exchangeability introduced from the summation constraint on multinomial parameters, and (3) the irreducibility of the multinomial logistic normal distribution that necessitates the use of parameter augmentation, or similar techniques, during inference. Using real and synthetic data we show that a variational autoencoder augmented with a fast isometric log-ratio (ILR) transform can address these issues and accurately estimate principal components from multinomially logistic normal distributed data. This model can be optimized on GPUs and modified to handle mini-batching, with the ability to scale across thousands of dimensions and thousands of samples.

## 2 INTRODUCTION

Many scientific disciplines that collect survey data, such as economics, psychology, political science and the biological sciences routinely deal with compositional data, where only relative information can be measured. These datasets are often in the form of counts, where the total counts within a sample are only indicative of the confidence of the measured proportions. The resulting proportions lie within a simplex and failing to account for the structure of this simplicial sample space can confound the interpretation of the measurements. As a result, there has been wide discussion across disparate disciplines (1; 2; 3; 4) concerning the reproducibility crisis that has arisen from the misinterpretation of compositional data. One of the obstacles to the appropriate analysis of compositional data is the difficulty of efficiently estimating the latent parameters that lie in the simplex.

Accurately scaling probabilistic inference across high-dimensional count data is a major outstanding challenge (5). This problem is apparent in the social sciences and is particularly pronounced in biological fields where datasets can obtain observations on tens of thousands of features across hundreds or millions of samples. One major computational bottleneck with Gaussian distributed data is the inversion of a $d$-dimensional covariance matrix that has a runtime of $O(d^3)$ (6; 7). As a result, probabilistic covariance estimation for high-dimensional data is a computationally challenging problem.

Recent theoretical developments (8) cementing the connection between Variational Autoencoders (VAEs) (9) and Probabilistic Principal Components Analysis (PPCA) (10) holds much promise for enabling accurate, scalable, low-rank approximations of large covariance matrices. Variational autoencoders were originally proposed as a generative model (9), but are now commonly deployed across scientific disciplines and have made contributions to single-cell RNA sequencing (11), microbiome modeling (12), protein modeling (13; 14; 15), natural language processing (16) and image processing (9). Following insights that connected regularized linear autoencoders and PCA (17), Lucas et al. (8) showed that carefully designed VAEs can recover the weights that are solved by PPCA. A computational advantage of VAEs is that they do not require the inversion of a covariance matrix, and the resulting runtime is $O(ndkT)$ for $n$ samples, $d$ dimensions, $k$ latent dimensions and $T$ epochs. While it has been noted that VAEs may take tens of thousands of epochs to estimate

the principal component (18), VAEs are easily parallelizable and can be accelerated with GPUs, presenting an attractive alternative to estimating principal components (17) and the resulting covariance matrix.

The connection between VAEs and PPCA is currently limited to Gaussian distributed data and not well-suited to a compositional setting. Showing that VAEs can recover the correct principal components from count data is nontrivial due to the non-conjugacy issues between the logistic normal distribution and count distributions such as the multinomial distribution. Furthermore, the parameters of the multinomial distribution are compositional; they are constrained within the simplex and the resulting covariance matrix is singular and non-invertible (1; 19). Aitchison (20) showed that PCA can be adapted to compositional data through the use of the center log-ratio (CLR) transform, which maintains isometry. However, this transformation is not isomorphic, requiring that the resulting log-ratios sum to zero, and as a result, CLR-transformed data will produce a singular covariance matrix and rank-deficient principal components. It has been shown that the isometric log-ratio (ILR) transform (21) satisfies both isomorphism and isometry and can handle this singularity issue (22; 23) while enabling the estimation of full-rank principal components. Here, we show that VAEs augmented with the ILR transform can infer principal components learned from PPCA on multinomially distributed data, beginning to address these critical shortcomings.

## 3 RELATED WORK

In the microbiome literature, there have been a number of methods (24; 25; 26; 27; 28) that have attempted to model ecological networks through the estimation of pairwise microbe correlations or pairwise inverse-covariance, where microbes are aggregated at different taxonomical scales or 'taxa'. Of these tools, only Flashweave can scale across more than thousands of taxa; however, it does this by avoiding the estimation of the covariance matrix. Methods that attempt to estimate the covariance matrix can only handle on the order of a few thousand dimensions. Although there is no widely accepted consensus definition of Multinomial PPCA in this context, being able to efficiently estimate the parameters of Multinomial PPCA would be highly useful for exploratory biological analysis. A number of studies have proposed using mixture modeling as a proxy for PCA (29; 30; 31); however, these techniques depend either on the Dirichlet distribution, whose covariance matrix is not flexible, or on stick-breaking, which violates permutation invariance (32).

Lucas et al. (8) has previously shown that the following two models can obtain the same maximum likelihood estimates of principal components $\boldsymbol{W}$:

$$
\begin{array}{c|c}
\text{Probabilistic PCA} & \text{Linear VAE} \\
p(\boldsymbol{x}|\boldsymbol{z}) = \mathcal{N}(\boldsymbol{W}\boldsymbol{z} + \boldsymbol{\mu}, \sigma^2 \boldsymbol{I}_d) & p(\boldsymbol{x}|\boldsymbol{z}) = \mathcal{N}(\boldsymbol{W}\boldsymbol{z} + \boldsymbol{\mu}, \sigma^2 \boldsymbol{I}_d) \\
p(\boldsymbol{z}) = N(\boldsymbol{0}, \boldsymbol{I}_k) & q(\boldsymbol{z}|\boldsymbol{x}) = \mathcal{N}(\boldsymbol{V}(\boldsymbol{x} - \boldsymbol{\mu}), \boldsymbol{D})
\end{array}
$$

Here, $p(\boldsymbol{x}|\boldsymbol{z})$ denotes the likelihood of observations $\boldsymbol{x} \in \mathbb{R}^d$ given the latent representation $\boldsymbol{z} \in \mathbb{R}^k$, $p(\boldsymbol{z})$ denotes the prior on $\boldsymbol{z}$ and $q(\boldsymbol{z}|\boldsymbol{x})$ denotes the estimated variational posterior distribution of $\boldsymbol{z}$ given an encoder parameterized by $\boldsymbol{V}$ and diagonal variances $\boldsymbol{D}$. Both models estimate the same low dimensional representation of the data through $\boldsymbol{z}$, and learn the same factorization of the covariance matrix through $\boldsymbol{W}$. While PPCA parameters are typically estimated through expectation maximization (10), linear VAEs are optimized by maximizing the Evidence Lower Bound (ELBO) given by

$$
\log p(\boldsymbol{x}) \geq \mathbb{E}_{q(\boldsymbol{z}|\boldsymbol{x})}\big[\log p(\boldsymbol{x}|\boldsymbol{z})\big] - KL\big(q(\boldsymbol{z}|\boldsymbol{x})||p(\boldsymbol{z})\big)
$$

For linear VAEs with a Gaussian likelihood, the variational posterior distribution $q(\boldsymbol{z}|\boldsymbol{x})$ can be shown to analytically agree with the posterior distribution $p(\boldsymbol{z}|\boldsymbol{x})$ learned from PPCA (8). However, deriving this connection for count-based likelihoods such as the multinomial distribution is complicated due to non-conjugacy issues (Appendix A). This is a major obstacle for many biological applications; multiple works have shown the merits of incorporating count distributions explicitly into the model (33; 34; 35; 36). Here, we provide directions for overcoming this issue.

## 4 METHODS

First, we will redefine Multinomial PPCA with the ILR transform (21). Then we will make the connection between Multinomial VAEs and Multinomial PPCA by leveraging insights from the

Collapse-Uncollapse (CU) sampler (33). We will then derive an algorithm to obtain the maximum a posteriori (MAP) estimate for the VAE parameters.

## 4.1 PROBABILISTIC MULTINOMIAL PCA

PPCA can be extended to multinomially distributed data with the following generative model:

$$p(\boldsymbol{x}|\boldsymbol{\eta}) = \text{Mult}(\phi(\boldsymbol{\Psi}\boldsymbol{\eta})) \tag{1}$$

$$p(\boldsymbol{\eta}|\boldsymbol{z}) = \mathcal{N}(\boldsymbol{W}\boldsymbol{z}, \sigma^2 \boldsymbol{I}_{d-1}) \tag{2}$$

$$p(\boldsymbol{z}) = \mathcal{N}(\boldsymbol{0}, \boldsymbol{I}_k) \tag{3}$$

Here $\boldsymbol{W} \in \mathbb{R}^{d-1 \times k}$ represents the PCA loading matrix, $\sigma^2$ is the variance, $\boldsymbol{\Psi} \in \mathbb{R}^{d \times d-1}$ is a fixed contrast matrix whose columns sum to zero and $\phi$ is the softmax transform. For a single sample, $\boldsymbol{x} \in \mathbb{N}^d$ are the observed $d$-dimensional counts, $\boldsymbol{\eta} \in \mathbb{R}^{d-1}$ are the latent logits and $\boldsymbol{z} \in \mathbb{R}^k$ is the latent representation. The term $\phi(\boldsymbol{\Psi}\boldsymbol{\eta})$ is distributed logistic normal, $\phi(\boldsymbol{\Psi}\boldsymbol{\eta}) \sim \mathcal{LN}(\boldsymbol{W}\boldsymbol{z}, \sigma^2 I)$, as shown by Aitchison (37). Furthermore, $p(\boldsymbol{x}|\boldsymbol{z})$ yields a multinomial logistic normal distribution, which is given by marginalizing out $\boldsymbol{\eta}$ in the following expression:

$$\mathcal{MLN}(\boldsymbol{x}|\boldsymbol{z}) = \int_{\boldsymbol{\eta}} p(\boldsymbol{x}|\boldsymbol{\eta})p(\boldsymbol{\eta}|\boldsymbol{z})d\boldsymbol{\eta}$$

This integral is not tractable; as a result, this distribution does not have an analytically defined expectation, variance or probability density function. There have been multiple attempts to estimate the posterior distribution with MCMC (38; 39; 35; 40), but the complexity of this distribution requires a large number of samples, limiting the scalability of these methods. Variational methods have been developed to estimate the logistic normal distribution, but due to conditional non-conjugacy, these methods often rely on approximations to the ELBO, further complicating estimation (41).

Recently, Silverman et al. (33) proposed to use a Laplace approximation to estimate the parameters of the multinomial logistic normal posterior distribution. This approach relies on a two-stage optimization procedure on the factorized posterior distribution given by

$$p(\boldsymbol{\eta}, \boldsymbol{z}|\boldsymbol{x}) \propto p(\boldsymbol{\eta}|\boldsymbol{x})p(\boldsymbol{z}|\boldsymbol{\eta}) \tag{4}$$

If $\boldsymbol{\eta}$ can be directly estimated, then conditional independence can be used to factorize the posterior distribution. Since the probability densities of the multinomial distribution and the normal distribution are both log-convex functions, a global optimum can be obtained for the multinomial logistic normal distribution (33) (Appendix B.3). Furthermore, the multinomial distribution does not introduce additional local optima for estimating Multinomial PCA. Given this, in addition to recent evidence that a PPCA MAP estimator can be obtained from regularized linear autoencoders (17), we can design a new algorithm to obtain a Multinomial PPCA MAP estimator.

## 4.2 THE ILR TRANSFORM ENFORCES IDENTIFIABLITY

The softmax function is a shift-invariant function, which introduces an identifiability issue that has been addressed by the compositional data analysis community (1; 42). In order to remove the identifiability issues, an isomorphism between the logits $\boldsymbol{\eta}$ and the multinomial parameters must be maintained. One commonly used solution is to use a degenerate softmax, also known as the inverse additive log-ratio (ALR) transform (1) (Appendix B.2). Previous work has suggested that the isometric log-ratio transform (ILR) (21; 22) is more suitable for principal components analysis (Appendix B). The ILR and inverse ILR are given as follows:

$$\text{ILR}(\boldsymbol{x}) = \boldsymbol{\Psi}^T \log \boldsymbol{x} \qquad \text{ILR}(\boldsymbol{x})^{-1} = \phi(\boldsymbol{\Psi}\boldsymbol{x}) \tag{5}$$

where $\boldsymbol{\Psi} \in \mathbb{R}^{d \times d-1}$ is a basis such that $\boldsymbol{\Psi}^T \boldsymbol{\Psi} = \boldsymbol{I}_{d-1}$ and $\boldsymbol{\Psi}\boldsymbol{\Psi}^T = \boldsymbol{I}_d - \frac{1}{d}\boldsymbol{1}_{d \times d}$. A naive implementation of the ILR transform can be memory-intensive and computationally intensive for large $d$. However, any orthonormal basis can be used to parameterize the ILR transform and some of these bases can be represented by binary trees (43; 44; 45). A binary tree can be used to represent $\boldsymbol{\Psi}$ with $O(d \log d)$ elements, where the $l$th column vector of $\boldsymbol{\Psi}$ is given as follows:

$$\boldsymbol{\Psi}_{\cdot l} = (\underbrace{0, \ldots 0}_{k}, \underbrace{a, \ldots a}_{r}, \underbrace{b, \ldots, b}_{s}, \underbrace{0, \ldots, 0}_{t}) \tag{6}$$

$$a = \frac{\sqrt{|\boldsymbol{s}|}}{\sqrt{|\boldsymbol{r}|(|\boldsymbol{r}| + |\boldsymbol{s}|)}} \qquad b = \frac{-\sqrt{|\boldsymbol{r}|}}{\sqrt{|\boldsymbol{s}|(|\boldsymbol{r}| + |\boldsymbol{s}|)}} \tag{7}$$

where $l$ indexes an internal node in the tree with left children $r$, right children $s$, nodes to the left $k$ and nodes to the right $t$ (46) (Figure S2). Due to rotation invariance, it doesn't matter which tree is used to parameterize the ILR basis, but the choice of tree can influence the runtime of the ILR transform. If a balanced binary tree is used, the memory requirements representing $\boldsymbol{\Psi}$ can be brought down from $O(d^2)$ to $O(d \log d)$ and can reduce the matrix vector multplication runtime from $O(d^2)$ to $O(d \log d)$ (See Appendix B.1). This can speed up the matrix-vector multiplication operations by an order of magnitude for datasets with more than ten thousand dimensions.

## 4.3 MULTINOMIAL VARIATIONAL AUTOENCODER ARCHITECTURE

Our full Multinomial VAE model is given as follows:

$$p(\boldsymbol{x}|\boldsymbol{\eta}) = \text{Mult}(\phi(\boldsymbol{\Psi}\boldsymbol{\eta})) \tag{8}$$

$$p(\boldsymbol{\eta}|\boldsymbol{z};\boldsymbol{\theta_{dec}}) = \mathcal{N}(\boldsymbol{W}\boldsymbol{z} + \boldsymbol{\mu}, \sigma^2 \boldsymbol{I_{d-1}}) \tag{9}$$

$$q(\boldsymbol{z}|\boldsymbol{x};\boldsymbol{\theta_{enc}}) = \mathcal{N}(F_L(\boldsymbol{\Psi}^T(\widetilde{\log}(\boldsymbol{x}) - \boldsymbol{\mu})), \boldsymbol{D}) \tag{10}$$

where $\boldsymbol{\theta_{dec}} = \{\boldsymbol{W}, \sigma^2\}$ denotes the decoder parameters, $\boldsymbol{\theta_{enc}} = \{F_L, \boldsymbol{D}\}$ denotes the encoder parameters and $\boldsymbol{\mu} \in \mathbb{R}^{d-1}$ is a bias parameter. Here, $q(\boldsymbol{z}|\boldsymbol{x};\boldsymbol{\theta_{enc}})$ denotes the variational posterior distribution of $\boldsymbol{z}$ given by the encoder represented as an $L$-layer dense neural network with appropriate activations. This encoder is directly used to evaluate $p(\boldsymbol{\eta}|\boldsymbol{z};\boldsymbol{\theta_{dec}})$. Furthermore, flat priors are assumed for all variables except $\boldsymbol{z}$.

It is important to note potentially challenging modeling issues when designing the encoder. The ILR transform is not directly applicable to count data, since $\log(0)$ is undefined. A common approach to this problem is to introduce a pseudocount before applying a logarithm, which we will denote as $\widetilde{\log}(\boldsymbol{x}) = \log(\boldsymbol{x} + 1)$. The choice of pseudocount is arbitrary and can introduce biases. To alleviate this issue, we introduce the deep encoder neural network highlighted in Equation 10; we expect that the universal approximation theorem would apply here (47; 48) and that the accuracy of estimating the latent representation $\boldsymbol{z}$ will improve with more complex neural networks. This is supported in our simulation benchmarks; more complex encoder architectures can better remove biases induced from the pseudocounts.

## 4.4 ALTERNATING MINI-BATCH OPTIMIZATION PROCEDURE

Given that our objective here is to obtain the MAP estimate of the VAE model parameters, the VAE parameters $\boldsymbol{\theta} = \{F_L, \boldsymbol{W}, \boldsymbol{D}, \sigma^2\}$ can be obtained by estimating the global maximum of the posterior distribution. In the original CU sampler implementation, the parameter $(\boldsymbol{\eta_1}, \ldots, \boldsymbol{\eta_n})$ is optimized across the entire dataset and then $(\boldsymbol{\eta_1}, \ldots, \boldsymbol{\eta_n})$ is fixed in order to estimate the remaining parameters. For large studies, this can be memory demanding, since $(\boldsymbol{\eta_1}, \ldots, \boldsymbol{\eta_n}) \in \mathbb{R}^{n \times d-1}$ alone can scale to millions of parameters.

To scale this estimation procedure to large high-dimensional datasets we have devised a mini-batched alternating minimization procedure. For a mini-batch $\boldsymbol{X}^{(i)} = (\boldsymbol{x}^{(1)}, \ldots, \boldsymbol{x}^{(b)})$ of size $b$

---

**Algorithm 1** VAE Alternating Maximization Optimization

> **repeat**
>     **for** $\boldsymbol{X}^{(i)} \in \boldsymbol{X}$ **do**
>         $\hat{\boldsymbol{H}} \leftarrow \arg\max\limits_{\boldsymbol{\eta}} \left[ \log p(\boldsymbol{X}^{(i)}|\boldsymbol{H}) \right] \in \mathbb{R}^{b \times d-1}$
>         $\boldsymbol{\theta} \leftarrow \arg\max\limits_{\boldsymbol{\theta}} \left[ \log q(\hat{\boldsymbol{H}}|\boldsymbol{X}^{(i)}; \boldsymbol{\theta}) + \log p(\boldsymbol{Z}) \right]$
>     **end for**
> **until** convergence

---

and corresponding latent variables $\boldsymbol{Z}^{(i)} = (\boldsymbol{z}^{(1)}, \ldots, \boldsymbol{z}^{(b)})$ and $\boldsymbol{H}^{(i)} = (\boldsymbol{\eta}^{(1)}, \ldots, \boldsymbol{\eta}^{(b)})$, the quantities $\log p(\boldsymbol{X}^{(i)}|\boldsymbol{H}^{(i)})$ and $\log p(\boldsymbol{Z}^{(i)})$ are given by Equation 1 and 3. The variational posterior

distribution of $\boldsymbol{\eta}$ given by $q(\boldsymbol{H}^{(i)}|\boldsymbol{X}^{(i)};\boldsymbol{\theta}) = \prod_{j=1}^{b} q(\boldsymbol{\eta}^{(j)}|\boldsymbol{x}^{(j)};\boldsymbol{\theta})$ can be obtained by marginalizing out $\boldsymbol{z}$ in Equations 8, 9 and 10 as follows:

$$q(\boldsymbol{\eta}^{(j)}|\boldsymbol{x}^{(j)};\boldsymbol{\theta}) = \mathcal{N}\big(\boldsymbol{W} F_L\big(\boldsymbol{\Psi}^T(\widetilde{\log}(\boldsymbol{x}^{(j)}) - \boldsymbol{\mu})\big) + \boldsymbol{\mu}, \ \boldsymbol{W}\boldsymbol{D}\boldsymbol{W}^T + \sigma^2 \boldsymbol{I}_d\big) \qquad (11)$$

The prior $\log p(\boldsymbol{Z}^{(i)})$ is evaluated, given $\log p(\boldsymbol{Z}^{(i)}) = \log \mathcal{N}(\boldsymbol{Z}^{(i)}|0, \boldsymbol{I}_k)$ and the latent encoding means given by $\boldsymbol{Z}^{(i)} = F_L\big(\boldsymbol{\Psi}^T(\widetilde{\log}(\boldsymbol{X}^{(i)}) - \boldsymbol{\mu})\big)$, where $\boldsymbol{Z}^{(i)}$ is integrated out of Equation 11.

## 5 RESULTS

### 5.1 MULTINOMIAL PPCA AND VAES AGREE IN SIMULATION BENCHMARKS

To determine if the proposed multinomial variational autoencoder can recover principal components, we extended the benchmarks proposed in (18). Here, we benchmarked our proposed analytical VAE

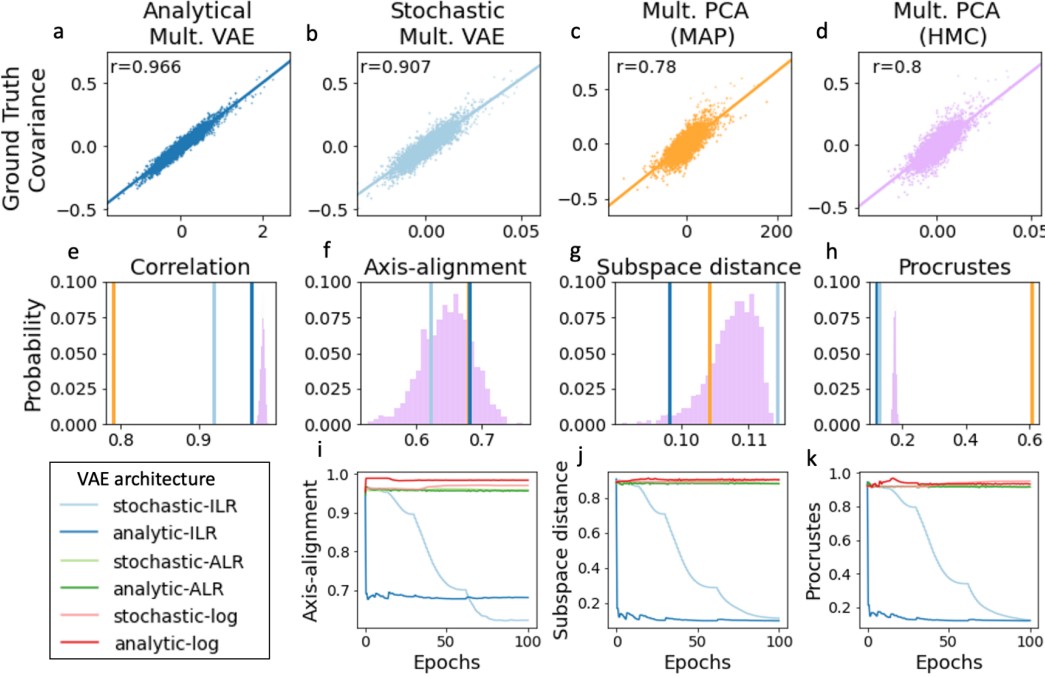

Figure 1: Dense simulation benchmarks. (a-d) Comparision of ground truth covariances (y-axis) and estimates from analytical VAE, stochastic VAE, MAP estimator and HMC performed in Stan (x-axes); all of these methods use the ILR transform and only the posterior means are shown in (d). (e-h) Correlation, axis-alignment, subspace distance and Procrustes metrics between analytical VAEs, stochastic VAEs, MAP and HMC after convergence. (i-k) Comparison of log, ALR and ILR transforms with respect to axis-alignment, subspace distance and Procrustes. The axis-alignment metric measures the angular differences between $\boldsymbol{W}$ and the ground truth principal components. The subspace distance is a measure of agreement between the ground truth correlations and the correlations estimated from $\boldsymbol{W}$. Procrustes measures the residual error after obtaining the best rotation and scaling factors to match $\boldsymbol{W}$ to the ground truth principal components. See Appendix C.5 for descriptions of the correlation, axis-alignment, subspace-distance and Procrustes metrics.

to the stochastic VAEs (9) with a multinomial likelihood across multiple simulations. These methodologies were compared against the ground truth covariances of the multinomial logistic normal, in addition to the MAP and Hamiltonian Monte Carlo (HMC) estimates obtained from Stan (49). The agreement between the ground truth principal components and the estimated principal components is measured by axis-alignment, subspace distance, correlation and Procrustes (50) (see Appendix C.4 for details).

When fitting against multinomial logistic normal distributed data with no zero counts, all of the ILR-based methodologies can accurately estimate the covariance matrix up to scale (Figure 1a-d). The analytical VAE MAP estimate and the posterior samples of HMC all have a correlation close to 1, suggesting a strong agreement between the ground truth covariance and the estimated covariances. If the principal components were perfectly estimated, the axis-alignment, subspace distance and Procrustes metrics would all be close to zero. While the subspace distance and Procrustes are notably close to zero, the axis-alignment metric is above 0.5, which would suggest disagreement between the ground-truth and the estimated principal component axes. Given that the posterior distribution of the axis-alignment metrics obtained from HMC overlaps with the analytical VAE, stochastic VAE and MAP estimates, there is evidence that reducing this metric to zero is inherently difficult for count data. However, both the subspace distance and Procrustes metrics approach zero, supporting our claim that principal components can be estimated up to scale and rotation on fully observed count data. Furthermore, our simulations (Figure 1i-k and Figure 2a-h) suggest that among the log-ratio transforms benchmarked, only the ILR transform can recover principal components, corroborating previous findings surrounding compositional PCA (42; 23). Dealing with sparse data presents addi-

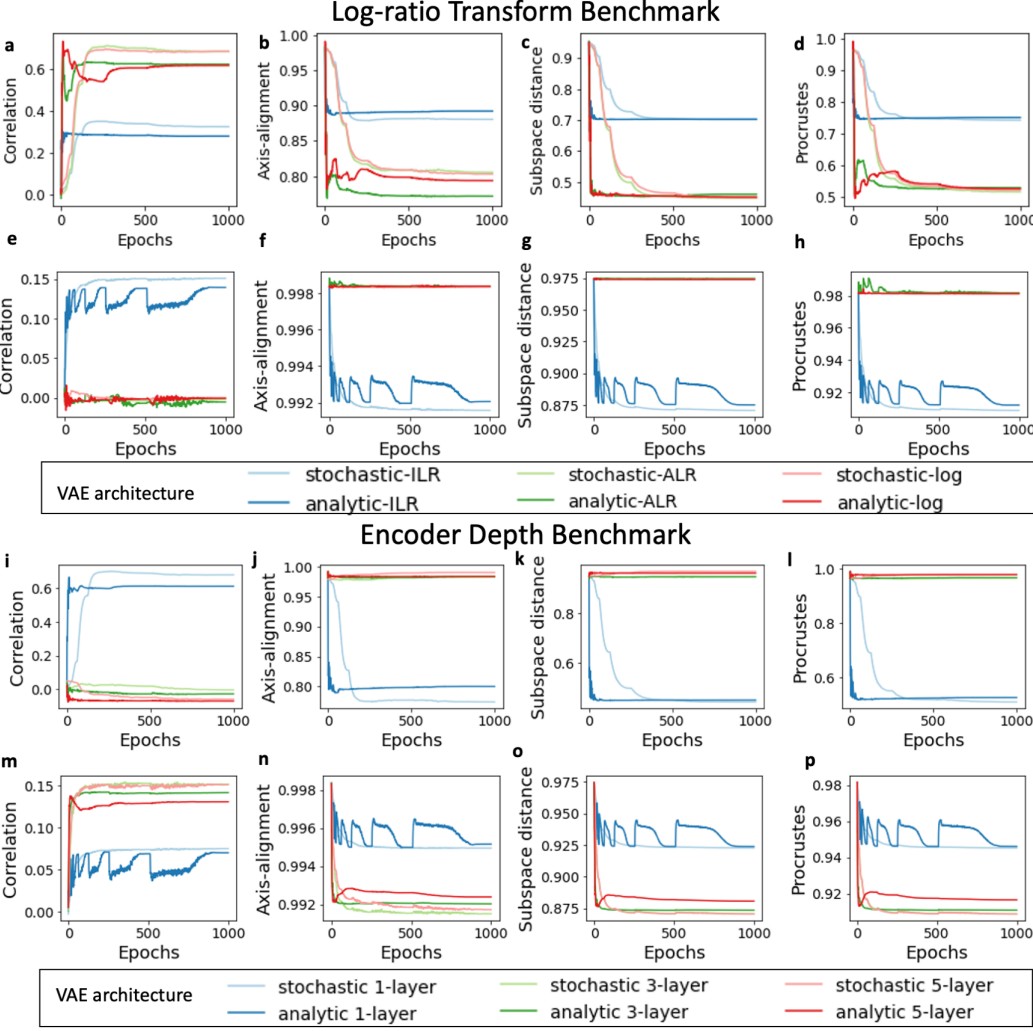

Figure 2: Comparison of different log-ratio transforms (a-f) and encoder depths (g-l) in sparse simulations. Two datasets with 200 and 5000 input dimensions are simulated for benchmarks (a-d, i-l) and (e-h, m-p) respectively. Log-ratio transform benchmarks investigate different log-ratio transforms using a single-layer encoder. The Encoder Depth benchmark investigates the impact of the number of encoder layers combined with the ILR transform.

tional challenges; the zeros in count data are indicative of missing data, which can further complicate the estimation of principal components. Our hypothesis that boosting the complexity of the encoder architecture would help alleviate issues with missing data is supported by benchmarks shown in Figure 2i-p. In both of the sparse datasets, none of the methods were able to achieve optimal accuracy across any of the benchmark metrics, but there is a clear advantage of utilizing multilayer encoders compared to single-layer encoders.

Across the simulation benchmarks, the analytical and stochastic VAEs have comparable performance, with discrepancies highlighted by the correlation metric in the dense and sparse benchmarks. On the dense datasets, our proposed analytical VAE has better agreement with the ground truth covariance metric, whereas on the sparse datasets, it appears that the stochastic VAE has better agreement. It is difficult to explain the reason for the performance gap between our proposed analytical VAE and the stochastic VAE due the analytical intractability of the Multinomial VAE ELBO. The challenge of accurately estimating an optimal $H$ for each mini-batch could be one limiting factor affecting the performance of our proposed analytical VAE (Appendix B.4).

## 5.2 PRETRAINING MULTINOMIAL VAES ON A VERY LARGE COMPENDIUM OF MOUSE MICROBIOME OBSERVATIONS

To showcase this on real microbiome datasets, 11k mouse fecal pellet samples with a total of 8.8k features across 45 studies were analyzed. We trained 2 models to evaluate the differences between the stochastic VAE and the analytical VAE. We trained these models for 10k epochs; the models with the smallest validation error are reported here. The details behind the full training procedure are given in Appendix C.3. Visualization of the learned decoder weights shown in Figure 3 makes

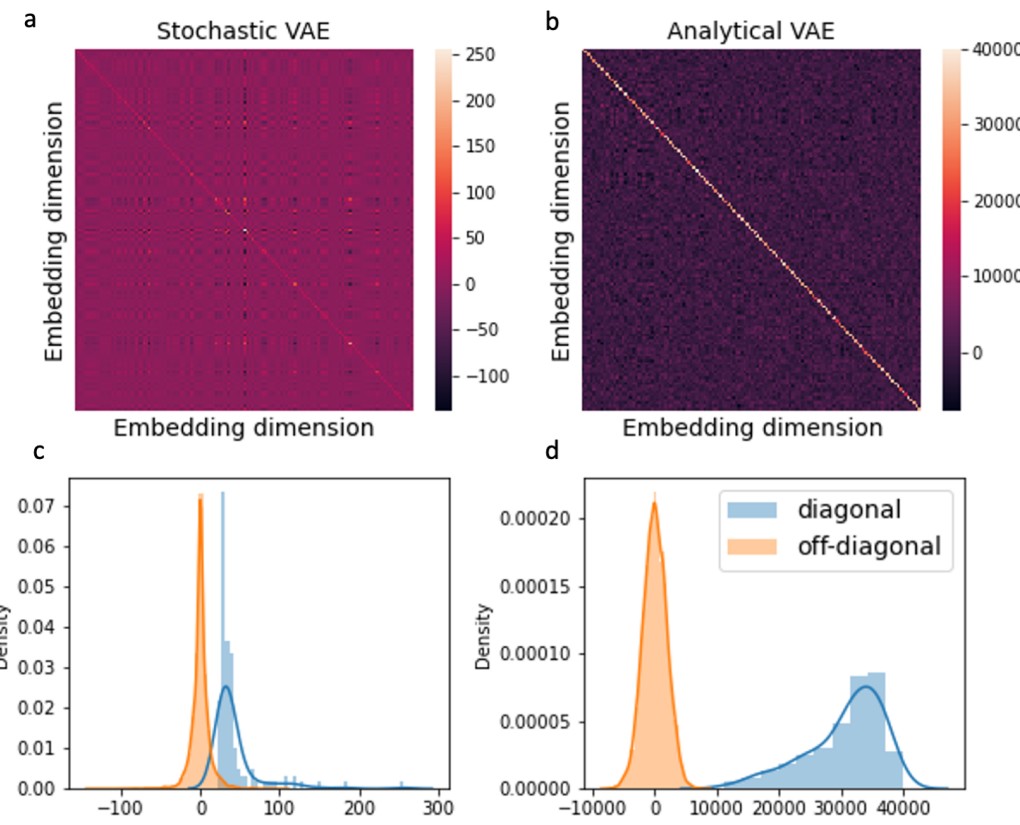

Figure 3: Comparison of the decoder inner product between the pretrained stochastic VAE and the pretrained analytical VAE. (a-b) Heatmap of $W^T W$. (c-d) Distribution of diagonal elements and off-diagonal elements of $W^T W$.

it clear that the analytical VAE and, to a lesser extent, the stochastic VAE are able to learn orthog-

onal embeddings, since $\boldsymbol{W}^T\boldsymbol{W}$ approximates a diagonal matrix. The orthogonality of the decoder weights is more apparent in our proposed analytical VAE than in the stochastic VAE decoder; there is a larger distinction between the diagonal elements and the off-diagonal elements. Unlike linear VAEs (8), the connection between the eigenvalues and the decoder weights is not as clear due to the asymmetry between the encoder and the decoder; scale identifiability of the decoder weights complicates the process of estimating the eigenvalues. This is apparent in Figure 3; heatmaps of the $\boldsymbol{W}$ inner products are on vastly different scales.

One of the advantages of employing a pretraining strategy with VAEs is that the pretrained models can enable one-shot learning (51). By estimating a low-dimensional representation from a large unlabeled high-dimensional dataset, fewer labeled samples for training downstream classifiers. We showcase this property in two classification benchmarks.

## 5.3  Classification and correlation benchmarks

To construct benchmarks on real biological datasets, we analyzed 16S sequencing data obtained from a study conducted by Tripathi et al. investigating hypoxia that used mouse fecal pellets (52) and a study conducted by Shalapour et al. investigating cancer effects on mice (53).

Due to the connections between Multinomial VAEs and compositional PCA (20; 22), we expect that, if our VAE is performing well, we will see that the row Euclidean distances of $\boldsymbol{\Psi W} \in \mathbb{R}^{d \times k}$ correspond to Lovell's un-normalized proportionality metric (54; 23) given by

$$\left\| (\boldsymbol{\Psi W})_i - (\boldsymbol{\Psi W})_j \right\|_2^2 \propto \mathrm{Var}\left( \log \frac{x_i}{x_j} \right) \tag{12}$$

where $\boldsymbol{x}$ refers to the observed proportions and $i$ and $j$ refer to the two features being compared. The proportionality metric has a simple interpretation; if this variance is small, that implies that the two features are highly co-occurring.

Here, only a relative relationship between the proportionality metric and the VAE decoder row distances can be stated due to scale identifiability issues inherent in Multinomial PPCA. Proportionality has been shown to be a more reliable metric for compositional data compared to standard correlation metrics such as Pearson and Spearman in single cell RNA sequencing and microbiome studies (3; 55; 56; 2), supporting the notion that our proportional comparison of our VAE decoder to Lovell's un-normalized proportionality metric will provide a sound perspective, as well as a means for interpreting the weights of the VAE.

We compared the estimated VAE embeddings with the proportionality metric to determine if this relationship holds empirically. The pairwise decoder distances are compared to the pairwise proportionality metrics of 300 selected features (Table 1 and Figure S5). The learned VAE representations are benchmarked using K-nearest neighbors (KNN) classification. KNN is applied to the learned VAE encodings across the two microbiome datasets to determine how well the classifiers can identify the mice that were induced with hypoxia (52) and classify mice based on their experimental group (53). These classification models are compared against two baseline models, namely KNN trained on LDA topic proportions (31), and KNN trained on raw counts.

From the classification benchmarks illustrated in Table 1, we can see that there is a large margin between the VAE models and the baseline models across the measured metrics. Both VAEs had a higher F1 score and AUC, suggesting that the learned representations can better separate the experimental groups. Part of this performance boost could be attributed to the differences between distance metrics. When the ILR transform is utilized, the Euclidean distance between the latent encodings approximates the Aitchison distance between proportions (57; 23), which measures relative differences instead of absolute differences. Our findings collaborate theoretical insights from compositional data analysis and empirical microbiome observations, where the Aitchison distance provided a substantial boost in KNN classification accuracy compared to other distance metrics (58; 59). Furthermore, the average negative log-likelihood score (NLL) is lower for both of the VAE models compared to LDA, suggesting that the VAE models generalized better on held out data than LDA. This decrease in the predictive log-likelihood could be attributed to the increased flexibility of the covariance matrix in the logistic normal distribution compared to the Dirichlet distribution.

There are a couple of notable discrepancies between the two Multinomial VAE models. The stochastic VAE appears to have superior classification performance and lower reconstruction error com-

| Dataset | Method | NLL | F1 score | AUC | Proportionality |
|---------|--------|-----|----------|-----|-----------------|
| Shalapour et al. | analytical-VAE | 57750 | $0.700 \pm 0.021$ | $\mathbf{0.910} \pm 0.013$ | **0.60** |
| | stochastic-VAE | **11615** | $\mathbf{0.732} \pm 0.023$ | $0.891 \pm 0.015$ | 0.30 |
| | LDA | 169384 | $0.507 \pm 0.034$ | $0.742 \pm 0.018$ | NA |
| | raw | NA | $0.406 \pm 0.039$ | $0.686 \pm 0.018$ | NA |
| Tripathi et al. | analytical-VAE | 117789 | $0.867 \pm 0.005$ | $0.873 \pm 0.004$ | **0.62** |
| | stochastic-VAE | **13815** | $\mathbf{0.926} \pm 0.004$ | $\mathbf{0.928} \pm 0.003$ | 0.28 |
| | LDA | 132091 | $0.718 \pm 0.006$ | $0.744 \pm 0.005$ | NA |
| | raw | NA | $0.700 \pm 0.007$ | $0.729 \pm 0.006$ | NA |

Table 1: Classification and Correlation benchmarks. KNN classification was performed with k=5 for all representations. The average negative log-likelihood (NLL) was evaluated on samples that were held out during the training of the VAEs. 100 rounds of 10-fold cross validation were applied to estimate average F1 score, AUC and standard errors. "Raw" indicates that no transformation was applied to the raw counts before performing KNN classification. The agreement between the VAE decoder weights and Lovell's proportionality metric is measured according to Pearson's $r$ on the log-transformed metrics.

pared to the analytical VAE. The exception to this is the HCC dataset (53), where the analytical VAE marginally outperforms the stochastic VAE in terms of AUC. However, the analytical VAE can learn more apparent orthogonal embeddings (Figure 3) and better agrees with Lovell's proportionality metric (Table 1). Only the analytical VAE was able to recover the log-linear relations between Lovell's proportionality and the VAE embedding distances, suggesting that it can more accurately learn biologically relevant correlations (Figure S5).

## CONCLUSION

Prior work aiming to build frameworks for probabilistic estimation of covariance matrices from count data have been largely limited to conjugate priors due to their tractability. However, these choices can lead to models with lower explanatory power due to the rigid structure of the resulting covariances. Further, the correct treatment of compositional data requires additional development in this context. For example, disregarding the simplicial sample space associated with compositional data and performing Pearson correlations on raw count data is common practice in scientific applications, but is a source of reproducibility issues (2). Due to the negative bias induced from the covariance on the observed count data, the estimated covariances will not agree with the ground truth covariances in the system of interest, a fact noted by Pearson in 1897 (60).

Adapting the logistic normal distribution in place of conventional conjugate priors provides a means to remedy the issue of inferring correlations on compositional data. The covariance matrix on ILR-transformed data can be interpreted using Lovell's proportionality metric, and can serve as a replacement for pairwise correlations. Since log-ratios are scale-invariant, the dependence on the total counts disappears, which is critical for assuring agreement between the relative quantities measured through count data and the absolute quantities in the system that aren't directly observable. While there is sound motivation to employ the logistic normal distribution in this context, its application has been limited due to the challenge of estimating these distributions. Here, we have provided a means to accurately and efficiently estimate these distributions and covariance matrices up to scale using the ILR transform. To this end, we have provided a proof-of-concept that Multinomial VAEs can learn the principal components obtained from Multinomial PPCA.

We have shown that fitting low-rank approximations using Multinomial VAEs can provide more explanatory representations than LDA while providing a means to obtain interpretable correlations. These methods can be used in one-shot learning and transfer learning settings, requiring fewer labeled samples for training and allowing for the use of pretrained models to extract features from smaller datasets. Given the vast number of scientific disciplines that collect compositional data, we anticipate that models such as these Multinomial VAEs will have a significant impact on many scientific applications.

## CODE AVAILABILITY

All of our software and analyses can be found on Zenodo at http://doi.org/10.5281/zenodo.4289004

## ACKNOWLEDGEMENTS

We want to acknowledge Pytorch (61), Pytorch-Lightning (62), the Biom format (63), Matplotlib (64), Scipy (65), Numpy (66) and Stan (49) for providing the software foundation that this work was built upon.

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

# Appendices

## A CHALLENGES IN DERIVATION OF AN ANALYTICAL MULTINOMIAL VAE ELBO

Recall that the generative model for Multinomial PPCA is given as follows:

$$p(\boldsymbol{x}|\boldsymbol{\eta}) = \text{Mult}(\boldsymbol{x}|\phi(\boldsymbol{\Psi}\boldsymbol{\eta})) \tag{13}$$

$$p(\boldsymbol{\eta}|\boldsymbol{z}) = \mathcal{N}(\boldsymbol{\eta}|\boldsymbol{W}\boldsymbol{z} + \boldsymbol{\mu}, \sigma^2 \boldsymbol{I}_{d-1}) \tag{14}$$

$$p(\boldsymbol{z}) = \mathcal{N}(\boldsymbol{z}|\boldsymbol{0}, \boldsymbol{I}_k) \tag{15}$$

With this in mind, we wish to estimate variational distributions $q(\boldsymbol{\eta}, \boldsymbol{z}|\boldsymbol{x}) = q(\boldsymbol{\eta}|\boldsymbol{z})q(\boldsymbol{z}|\boldsymbol{x})$ to approximate the posterior $p(\boldsymbol{\eta}, \boldsymbol{z}|\boldsymbol{x})$. These variational distributions can both be chosen to be normal distributions as follows:

$$q(\boldsymbol{z}|\boldsymbol{x}) = \mathcal{N}(\boldsymbol{V}\boldsymbol{\Psi}^T(\widetilde{\log}(\boldsymbol{x}) - \boldsymbol{\mu}), \boldsymbol{D})$$

$$q(\boldsymbol{\eta}|\boldsymbol{z}) = p(\boldsymbol{\eta}|\boldsymbol{z})$$

Noting that $\boldsymbol{z} \sim \mathcal{N}(\boldsymbol{V}\boldsymbol{\Psi}^T(\widetilde{\log}(\boldsymbol{x}) - \boldsymbol{\mu}), \boldsymbol{D})$, $q(\boldsymbol{\eta}|\boldsymbol{x})$ can be derived from $q(\boldsymbol{z}|\boldsymbol{x})$ as follows:

$$q(\boldsymbol{\eta}|\boldsymbol{x}) = \mathcal{N}\big(\boldsymbol{W}\boldsymbol{V}\boldsymbol{\Psi}^T(\widetilde{\log}(\boldsymbol{x})) - \boldsymbol{\mu}) + \boldsymbol{\mu}, \boldsymbol{W}\boldsymbol{D}\boldsymbol{W} + \sigma^2 \boldsymbol{I}\big)$$

To fine-tune these variational distributions to approximate the posterior distribution, we can minimize the following KL divergence:

$$\underset{q(\boldsymbol{\eta}, \boldsymbol{z}|\boldsymbol{x})}{\arg\max} KL(q(\boldsymbol{\eta}, \boldsymbol{z}|\boldsymbol{x})||p(\boldsymbol{\eta}, \boldsymbol{z}|\boldsymbol{x}))$$

Since we cannot optimize this quantity directly, we opt instead to maximize the evidence lower bound (ELBO) given by

$$\mathbb{E}_{q(\boldsymbol{\eta}, \boldsymbol{z}|\boldsymbol{x})}\left[\log \frac{p(\boldsymbol{\eta}, \boldsymbol{z}|\boldsymbol{x})}{q(\boldsymbol{\eta}, \boldsymbol{z}|\boldsymbol{x})}\right] \geq \mathbb{E}_{q(\boldsymbol{\eta}, \boldsymbol{z}|\boldsymbol{x})}\left[\log \frac{p(\boldsymbol{x}|\boldsymbol{\eta})p(\boldsymbol{\eta}|\boldsymbol{z})p(\boldsymbol{z})}{q(\boldsymbol{\eta}, \boldsymbol{z}|\boldsymbol{x})}\right]$$

We can partition this lower bound into three parts, given as follows:

$$= \underbrace{\mathbb{E}_{q(\boldsymbol{\eta}|\boldsymbol{x})q(\boldsymbol{z}|\boldsymbol{x})}[\log p(\boldsymbol{x}|\boldsymbol{\eta})]}_{(i)} + \underbrace{\mathbb{E}_{q(\boldsymbol{\eta}|\boldsymbol{x})}\left[\log \frac{p(\boldsymbol{\eta}|\boldsymbol{z})}{q(\boldsymbol{\eta}|\boldsymbol{x})}\right]}_{(ii)} + \underbrace{\mathbb{E}_{q(\boldsymbol{z}|\boldsymbol{x})}\left[\log \frac{p(\boldsymbol{z})}{q(\boldsymbol{z}|\boldsymbol{x})}\right]}_{(iii)}$$

Since $\text{Mult}(\boldsymbol{x}|\boldsymbol{p}) \propto \sum_{i=1}^{d} x_i \log p_i$, the first term is given by

$$\mathbb{E}_{q(\boldsymbol{\eta}|\boldsymbol{x})q(\boldsymbol{z}|\boldsymbol{x})}\big[\log p(\boldsymbol{x}|\boldsymbol{\eta})\big] \propto \mathbb{E}_{q(\boldsymbol{\eta}|\boldsymbol{x})}\left[\sum_{i=1}^{d} x_i \phi(\boldsymbol{\Psi}\eta)_i\right]$$

$$\propto \int_{\eta} \mathcal{N}\big(\boldsymbol{W}\boldsymbol{V}\boldsymbol{\Psi}^T(\widetilde{\log}(\boldsymbol{x}) - \boldsymbol{\mu}), \boldsymbol{W}\boldsymbol{D}\boldsymbol{W} + \sigma^2 \boldsymbol{I}\big) \sum_{i=1}^{d} x_i \phi(\boldsymbol{\Psi}\boldsymbol{\eta})_i \, d\boldsymbol{\eta}$$

Estimating the above integral is equivalent to estimating the expectation of a logistic normal distribution, which does not have an analytical solution (37). As a result, the analytical ELBO for the Multinomial VAE is intractable.

## B THE ILR TRANSFORM

As outlined in Equation 5, any orthogonal basis $\mathbf{\Psi}$ can be used to perform the ILR transform. However, there are a select few bases that can be represented by a binary tree. To see how an orthogonal basis can be constructed from a binary tree, consider the illustration in Figure S1.

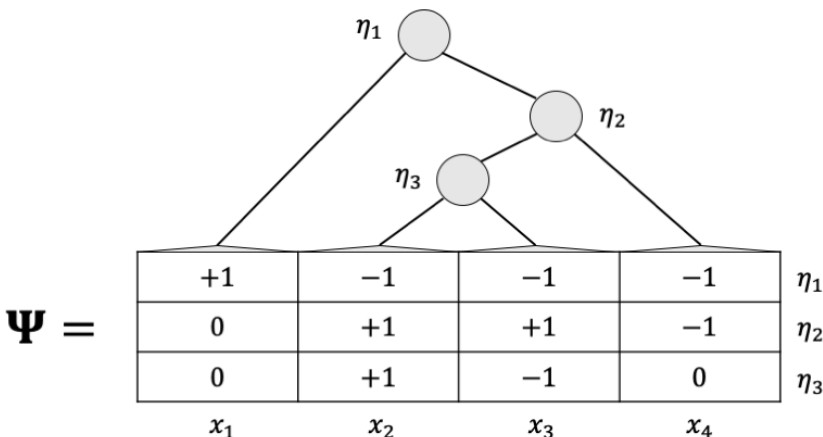

Figure S1: A small example showing how an orthogonal basis can be constructed from a binary tree. $\boldsymbol{x} = (x_1, x_2, x_3, x_4) \in \mathbb{S}^4$ represents species proportions and $\boldsymbol{\eta} = (\eta_1, \eta_2, \eta_3) \in \mathbb{R}^3$ represents the log-ratios in the internal nodes. $\mathbf{\Psi}$ is a matrix of orthogonal contrasts that also can be represented from a binary tree.

Here the rows of the matrix in Figure S1 are orthogonal and the resulting product $\boldsymbol{\eta} = \mathbf{\Psi}^T \log \boldsymbol{x}$ yields

$$\eta_1 = \log \frac{x_1}{x_2 x_3 x_4} \qquad \eta_2 = \log \frac{x_2 x_3}{x_4} \qquad \eta_3 = \log \frac{x_3}{x_4}$$

The contrast matrix $\mathbf{\Psi}$ can be forced to be orthonormal such that $\mathbf{\Psi}^T \mathbf{\Psi} = \boldsymbol{I}_{d-1}$, as highlighted in Equation 6. Furthermore, this construction can be scaled to large binary trees as shown in Figure S2. Here, $\boldsymbol{\eta}_l$ represents the log-ratios at the internal node $l$, given by

$$\eta_l = \sqrt{\frac{|\boldsymbol{r}||\boldsymbol{s}|}{|\boldsymbol{r}| + |\boldsymbol{s}|}} \log \frac{g(\boldsymbol{x_r})}{g(\boldsymbol{x_s})} \tag{16}$$

The runtime of this operation is further discussed in Appendix B.1.

### B.1 THE RUNTIME OF THE ILR TRANSFORM

The naive runtime of the ILR transform of a single sample $\boldsymbol{x} \in \mathbb{R}^d$ is $O(d^2)$ due to the running time of dense matrix-vector multiplication. As shown in Equation 5, the ILR transform can also be represented by log-linear transformation with a contrast matrix. The binary tree can be used to represent a contrast matrix, as discussed in (46).

If the binary tree is balanced, each row of $\mathbf{\Psi}$ will have $O(\log d)$ non-zero elements, since the tree has a height of $O(\log d)$. Given that there are $d$ rows, the matrix-vector multiplication behind the inverse ILR transform $\mathbf{\Psi}\boldsymbol{\eta}$ can be done in $O(d \log d)$. For the same reason, the ILR transform has a runtime of $O(d \log d)$.

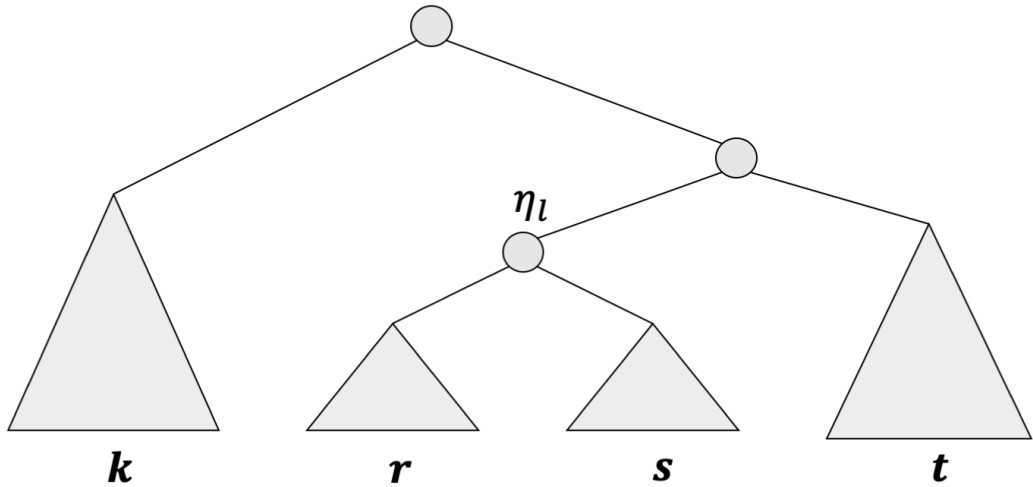

Figure S2: An illustration of how the ILR basis can be constructed on large trees. The quantities $g(\boldsymbol{x_r})$ and $g(\boldsymbol{x_s})$ yield the geometric means within a vector of proportion $\boldsymbol{x}$ for subsets $\boldsymbol{x_r}$ and $\boldsymbol{x_s}$. Here, $\boldsymbol{r}$ and $\boldsymbol{s}$ refer to the sets of features in the left and right subtrees for the internal node $l$. The log-ratios $\boldsymbol{\eta}$ can be obtained from either Equation 6 or Equation 16
.

## B.2 SHIFT INVARIANCE OF THE SOFTMAX TRANSFORM

There are two different scale identifiability issues. The softmax transform that is commonly used is shift invariant, where

$$\phi(\boldsymbol{x}) \leftarrow \phi(\boldsymbol{x} + a) \quad \forall a \in \mathbb{R}$$

This shift invariance will cause an identifiability issue when identifying the decoder matrix $\boldsymbol{W}$. The inverse ALR transform resolves this issue by enforcing an isomorphism; one of the coordinates is set to zero as follows:

$$\text{ALR}(\boldsymbol{x}) = \left[ \log \frac{x_1}{x_d}, \ldots, \log \frac{x_{d-1}}{x_d}, 0 \right], \quad \text{ALR}^{-1}(\boldsymbol{x}) = \phi\big((x_1, \ldots, x_{d-1}, 0)\big)$$

One of the implicit constructions of this transform is that the resulting contrast matrix is not orthogonal (42). Furthermore, because the ALR transform is not isometric, the resulting Euclidean distances in $z$ will not approximate the Aitchison distance. The ILR transform provides the best of both worlds, enforcing both isometry and isomorphism. Furthermore, there is a tight connection between the ILR transform and compositional PCA that is further discussed in the next section.

## B.3 GLOBAL LOG-CONVEXITY OF THE MULTINOMIAL LOGISTIC NORMAL DISTRIBUTION

Since the multinomial logistic normal distribution is difficult to directly evaluate, it is challenging to make statements regarding its maximum likelihood estimator. With the posterior factorization highlighted in Equation 4, we can make concrete statements about the original posterior factors. The log probability density of the multinomial distribution $p(x|\eta)$ can be written as

$$\log p(\boldsymbol{x}|\boldsymbol{\eta}) \propto \sum_{i=1}^{d} x_i \, \log \phi(\boldsymbol{\Psi}\boldsymbol{\eta})_i$$

$$= \sum_{i=1}^{d} x_i(\boldsymbol{\Psi}\boldsymbol{\eta})_i - m \log \left( \sum_{j=1}^{d} \exp(\boldsymbol{\Psi}\boldsymbol{\eta})_i \right)$$

$$= g(\boldsymbol{\eta}) \cdot T(\boldsymbol{x}) - A(\boldsymbol{\eta})$$

where $m = \sum_{i=1}^{d} x_i$ is the total number of counts and the functions $g(\boldsymbol{\eta})_i = (\boldsymbol{\Psi}\boldsymbol{\eta})_i$, $T(\boldsymbol{x})_i = x_i$, and $A(\boldsymbol{\eta}) = \log \left( \sum_{j=1}^{d} \exp(\boldsymbol{\Psi}\boldsymbol{\eta})_i \right)$ are the natural parameters of the exponential family distribu-

tion. The Hessian of $\log p(x|\eta)$ is given by

$$\frac{d^2 \log p(\boldsymbol{x}|\boldsymbol{\eta})}{d\eta_i d\eta_j} = \frac{d^2 A(\boldsymbol{\eta})}{d\eta_i d\eta_j}$$

Since $A(\boldsymbol{\eta})$ is strictly convex, $\log p(\boldsymbol{x}|\boldsymbol{\eta})$ is also strictly convex.

Similarly, the log probability density of the multivariate Gaussian distribution $p(\boldsymbol{z}|\boldsymbol{\eta}) = \mathcal{N}(\boldsymbol{\mu}, \boldsymbol{\sigma})$ is also strictly convex with respect to $\boldsymbol{\mu}$ and $\boldsymbol{\Sigma}$. Since the multinomial logistic normal distribution can be written as the sum of $\log p(\boldsymbol{z}|\boldsymbol{\eta})$ and $\log p(\boldsymbol{x}|\boldsymbol{\eta})$, it is also strictly convex. Therefore, there must be a unique optimal estimate for $\boldsymbol{\mu}$ and $\boldsymbol{\Sigma}$.

## B.4 MULTINOMIAL VARIATIONAL AUTOENCODER ESTIMATION

The covariance matrix $\boldsymbol{W}\boldsymbol{D}\boldsymbol{W}^T + \sigma^2 \boldsymbol{I}_{d-1} \in \mathbb{R}^{d-1 \times d-1}$ in Equation 11 can be efficiently inverted using the Woodbury identity (67; 68). The prior $\log p(\boldsymbol{Z^{(i)}})$ is evaluated given $\log p(\boldsymbol{Z^{(i)}}) = \mathcal{N}(\boldsymbol{Z^{(i)}}|0, \boldsymbol{I}_k)$, where $\boldsymbol{Z^{(i)}} = F_L\big(\boldsymbol{\Psi}^T(\widetilde{\log}(\boldsymbol{X}^{(i)}) - \boldsymbol{\mu})\big)$.

The optimal $\hat{\boldsymbol{H}}^{(i)}$ that maximizes $\log p(\boldsymbol{X}^{(i)}|\boldsymbol{H}^{(i)})$ is obtained through gradient descent optimization. Once $\hat{\boldsymbol{H}}^{(i)}$ is obtained, the remaining VAE parameters $\boldsymbol{\theta}$ can be estimated via gradient descent optimization. Like the EM algorithm proposed in (69), this procedure alternates between estimating $\boldsymbol{H}^{(i)}$ and $\boldsymbol{\theta}$ and repeats until convergence. For a single-layer linear encoder, this optimization procedure will eventually reach the global maxima with respect to $\boldsymbol{H}^{(i)}$ and the multivariate normal mean and covariance in $q(\boldsymbol{H}^{(i)}|\boldsymbol{\theta}, \boldsymbol{X}^{(i)})$, due to the log-convexity of the multinomial and normal distributions (Appendix B.3). On fully observed data, $\boldsymbol{W}$ and $\boldsymbol{D}$ can be estimated up to rotation and scale (Appendix B.6). This is not guaranteed for sparse count data, but using simulations, we can empirically show that increasing the complexity encoder architectures can help. Since $\hat{\boldsymbol{H}}^{(i)}$ needs to be accurately estimated before optimizing $F_L, \boldsymbol{W}, \boldsymbol{\sigma}$, and $\boldsymbol{D}$, multiple gradient descent steps are required for a given mini-batch.

In practice, obtaining the optimal $\hat{\boldsymbol{H}}^{(i)}$ for a given mini-batch $i$ may require hundreds of gradient descent steps from a random initialization. Since $\boldsymbol{H}^{(i)}$ is used to approximate the multinomial parameters describing the observed counts $\boldsymbol{x}$, we can initialize $\boldsymbol{H}^{(i)}$ with $\hat{\boldsymbol{H}}_0^{(i)} = \boldsymbol{\Psi}^T \widetilde{\log}(\boldsymbol{X}^{(i)})$. In practice, this can greatly reduce the number of gradient descent updates needed per mini-batch.

## B.5 THE CONNECTION BETWEEN MULTINOMIAL VAES AND COMPOSITIONAL PCA

With singular value decomposition, the resulting factors can be used to approximate the row and column distances. For a singular value decomposition given by $\boldsymbol{X} = \boldsymbol{U}\boldsymbol{S}\boldsymbol{V}^T$, row distances and column distances can be approximated as follows:

$$\|\boldsymbol{x}_{i.} - \boldsymbol{x}_{j.}\|_2 \approx \|s_i \boldsymbol{u}_i - s_j \boldsymbol{u}_j\|_2$$
$$\|\boldsymbol{x}_{.i} - \boldsymbol{x}_{.j}\|_2 \approx \|s_i \boldsymbol{v}_i - s_j \boldsymbol{v}_j\|_2$$

This relationship also yields a connection to the row and column covariances; the row covariances are given by $\boldsymbol{X}^T\boldsymbol{X} = \boldsymbol{U}\boldsymbol{S}^2\boldsymbol{U}$ and the column covariances are given by $\boldsymbol{X}\boldsymbol{X}^T = \boldsymbol{V}\boldsymbol{S}^2\boldsymbol{V}$.

A similar relationship applies to compositional PCA, except the singular value decomposition is applied to CLR-transformed values of $X$. The CLR transform is given by

$$\text{CLR}(\boldsymbol{x}) = \left[\log \frac{x_1}{g(\boldsymbol{x})}, \ldots, \log \frac{x_d}{g(\boldsymbol{x})}\right] \qquad \text{CLR}^{-1}(\boldsymbol{x}) = \phi(\boldsymbol{x})$$

The inverse CLR transform is equivalent to the softmax transform $\phi(\boldsymbol{x})$. The Euclidean distance on CLR and ILR transformed values is given by the Aitchison distance (57). As a result, the row and column distances are given by the Aitchison distance. The covariance matrix of CLR-transformed values is singular, and as a result, the resulting singular value decomposition will have a full rank of $d-1$. As shown by Egozcue et al. (23), the top $d-1$ components of the singular value decomposition are all in ILR coordinates. This observation cements the connection between compositional PCA and our proposed Multinomial VAE; since all of the singular value components can be represented

in ILR coordinates, this theoretically justifies the use of the ILR transform within our proposed Multinomial VAE.

Due to this connection, the distances between the learned VAE representations $z$ across pairs of samples should be proportional to the Aitchison distance. Furthermore, the distances between rows of $\mathbf{\Psi} W$ would also be given by the Aitchison distance, which is equivalent to Lovell's un-normalized proportionality constant (23).

### B.6    LACK OF SCALE IDENTIFIABILITY OF PPCA

As discussed in (8), VAEs will have the same identifiability issues that PPCA has; namely, for a diagonal matrix $A \in \mathbb{R}^{k \times k}$, the following equivalences hold:

$$W \leftarrow WA, \qquad V \leftarrow A^{-1}V$$

As a result, the decoder weights $W$ can only be identified up to scale. Furthermore, the Multinomial VAE architecture is asymmetric and, as a result, the Tranpose Theorem proposed by (17) is not as readily applicable. The lack of symmetry between $W$ and $V$ complicates the process of obtaining eigenvalues from $W$.

## C    SIMULATION AND MICROBIOME BENCHMARKS

### C.1    SIMULATION DETAILS

In the dense simulation highlighted in Figure 1, there were 100 features, 1000 samples and 1 million counts per sample. These counts were drawn from a multinomial logistic normal distribution whose covariance matrix had a rank of 10. The sparse dataset highlighted in Figure 2a-h contained 200 dimensions and 1000 samples, each of which contained 100 counts that were sampled from a multinomial logistic normal distribution whose covariance matrix had a rank of 10. The sparse dataset highlighted in Figure 2i-p contained 5000 dimensions and 10000 samples, each of which contained only 1000 counts total. These counts were sampled from a multinomial logistic normal distribution whose covariance matrix had a rank of 128.

The encoder architecture used for the stochastic and analytical VAEs consisted of fully connected dense layers with interwoven softplus activation functions. The intermediate encoder layers all have the same input and output dimension as the number of latent dimensions specified in the model. The same architecture was used for both the stochastic VAEs and the analytical VAEs. A schematic of the architecture is shown in Figure S3.

Multinomial PCA was fitted using both penalized likelihood estimation and Hamiltonian Monte Carlo using Stan (Appendix C.4). The initialization of these models was determined by the estimated factor loadings from the analytical VAE. This is particularly critical for fitting HMC on high-dimensional datasets. We were not able to run Stan on the sparse datasets.

### C.2    EXPERIMENTAL DATASET DETAILS

The full 8.8k mouse pellet dataset was retrieved from qiita (70) using redbiom (71).

In the hypoxia study conducted by Tripathi et al., mice were placed in simulated conditions to induce intermittent hypoxic/hypercapnic (IHH) stress, where the mouse's oxygen supply was reduced and its $CO_2$ supply was increased. The goal of this experiment was to detect the gut microbiome differences between the IHH mice and the control mice. We will use this information as a classification benchmark to determine if the experimental conditions the mice were placed under can be predicted from 16S sequencing counts. In total, there were 48 mice; 24 of these mice were placed in these IHH conditions and the remaining 24 mice served as controls. Fecal samples were collected twice a week for 6 weeks, resulting in 579 samples total. There were 5775 microbial taxa that were detected in this dataset.

The goal of the hepatocellular carcinoma (HCC) study conducted by Shalapour was to investigate the interplay between immunity, diet and carcinoma. In total, there were 478 mice with 52 phenotypes, split according to diet, immunity status and whether or not the mice were induced with HCC. Our

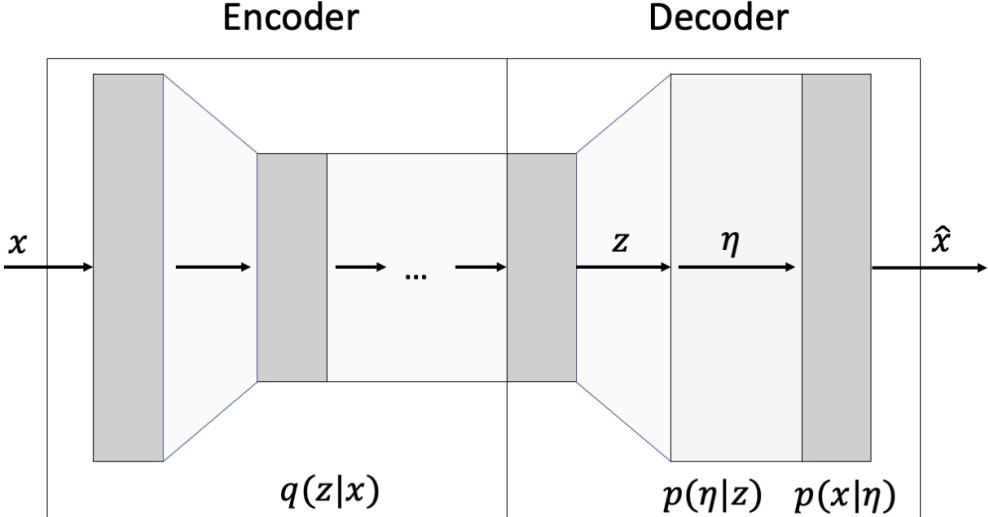

Figure S3: Visualization of VAE architecture

classification objective is to predict the mouse phenotype from the 16S sequencing counts obtained from their fecal pellets. In total, there was one fecal pellet collected from each mouse and there were 2794 microbial taxa detected across these samples. This benchmark is more challenging than the IHH benchmark due to the extreme class imbalance; some phenotypes contain upwards of 30 mice and other contain as few as 2 mice (Figure S4). The resulting classification benchmarks are shown in Table 1.

### C.3 MICROBIOME VAE TRAINING DETAILS

For the full 11k fecal pellet dataset, we focused on samples that were processed using closed-reference OTU picking using vsearch (72). This dataset was split into a 80/10/10 train/validation/test split in order measure out-of-distribution generalizability.

We trained our proposed analytical VAE and the stochastic VAE with 128 latent dimensions and a batch size of 1000 samples for 10000 epochs, with a learning rate of $10^{-3}$ and 100 gradient descent steps per mini-batch. Checkpoints were recorded every epoch; the models with the best validation error are reported. Cosine annealing with warm restarts was used as a learning rate scheduler, with the intention of easily escaping saddle points during optimization.

### C.4 METRICS FOR EVALUTING ESTIMATED DECODER WEIGHTS

To measure the agreement between the ground truth simulations and the estimated decoder weights $\boldsymbol{W}$, three different metrics were evaluated, namely, axis alignment and subspace distance, as proposed in (18), in addition to Procrustes analysis (50). Given the decoder weights $\boldsymbol{W}$ and the ground truth principal components $\boldsymbol{U}$, these metrics are defined as follows:

**Axis alignment**

$$d(\boldsymbol{U}, \boldsymbol{W}) = 1 - \frac{1}{k} \sum_{i=1}^{k} \max_{j} \frac{(\boldsymbol{U}_i^T \boldsymbol{W}_j)^2}{\|\boldsymbol{U_i}\|_2^2 \|\boldsymbol{W_j}\|_2^2}$$

This metric is ultimately a measure of the average cosine distance between the ground truth eigenvectors and the estimated decoder weights.

**Subspace distance**

$$d(\boldsymbol{U}, \boldsymbol{W}) = 1 - \text{Tr}(\boldsymbol{U}\boldsymbol{U}^T\boldsymbol{W}_*\boldsymbol{W}_*^T)$$

where $\boldsymbol{W}_*$ denotes the left singular vectors of $\boldsymbol{W}$, given by $\boldsymbol{W} = \boldsymbol{W}_*\boldsymbol{\Lambda}\boldsymbol{V}$. Since $\boldsymbol{U}\boldsymbol{U}^T$ yields the ground truth correlations and $\boldsymbol{W}_*\boldsymbol{W}_*^T$ yields the estimated correlations, this metric can be interpreted as a measure of agreement between the ground truth correlations and the estimated correlations.

**Procrustes Analysis**

$$d(\boldsymbol{U}, \boldsymbol{W}) = \underset{\boldsymbol{R},\boldsymbol{A}}{\arg\min}\||\boldsymbol{U} - \boldsymbol{W}\boldsymbol{R}\boldsymbol{A}\||_2^2$$

for $\boldsymbol{A}, \boldsymbol{R} \in \mathbb{R}^{k \times k}$, where $\boldsymbol{A}$ is a diagonal matrix and $\boldsymbol{R}$ is a rotation matrix. Prior to evaluating this metric, both $\boldsymbol{U}$ and $\boldsymbol{W}$ are standardized such that $\text{Tr}(\boldsymbol{U}\boldsymbol{U}^T) = \text{Tr}(\boldsymbol{W}\boldsymbol{W}^T) = 1$ and $\boldsymbol{W}$ is centered around the origin.

**Correlation**

$$d(\boldsymbol{U}, \boldsymbol{W}) = Corr(\text{vec}(A_{\boldsymbol{U}}), \text{vec}(A_{\boldsymbol{W}}))$$

where $A_{\boldsymbol{U}}(i, j) = \||\boldsymbol{u}_i - \boldsymbol{u}_j\||_2$ and $A_{\boldsymbol{W}}(i, j) = \||\boldsymbol{w}_i - \boldsymbol{w}_j\||_2$ denote the pairwise distances of $\boldsymbol{U}$ and $\boldsymbol{W}$. The pairwise distances are rotation invariant. Furthermore, the measure of the correlation is agnostic to scale, and will ignore the eigenvalue scale identifiability issues highlighted in the main text.

## C.5    STAN MULTINOMIAL PCA IMPLEMENTATION

```
data {
  int<lower=0> N;        // number of samples
  int<lower=0> D;        // number of dimensions
  int<lower=0> K;        // number of latent dimensions
  matrix[D-1, D] Psi;    // Orthonormal basis
  int y[N, D];           // observed counts
}

parameters {
  matrix[N, D-1] eta;  // ilr transformed abundances
  matrix[D-1, K] W;
  real<lower=0> sigma;
}

transformed parameters {
  matrix[D-1, D-1] Sigma;
  matrix[D-1, D-1] I;
  vector[D-1] z;
  I = diag_matrix(rep_vector(1.0, D-1));
  Sigma = W * W' + square(sigma) * I;
  z = rep_vector(0, D-1);
}

model {
  // generating counts
  for (n in 1:N){
     eta[n] ~ multi_normal(z, Sigma);
     y[n] ~ multinomial(softmax(to_vector(eta[n] * Psi)));
  }
}
```

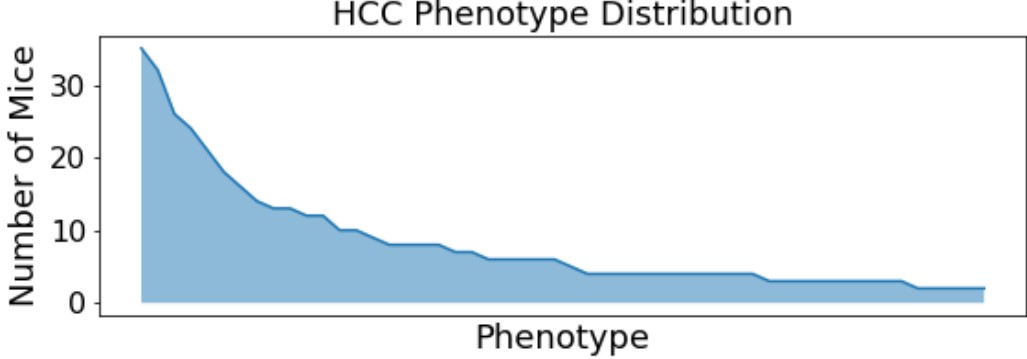

Figure S4: Distribution of 52 phenotypes in the Shalapour et al. study, highlighting the class imbalance inherent in this benchmark.

| Dataset | Method corrected | Pred Log prob | F1 score | AUC |
|---|---|---|---|---|
| Shalapour et al. | analytical-VAE | 57750 | 0.845 | 0.820 |
| | stochastic-VAE | 11615 | 1.000 | 1.000 |
| | LDA | 169384 | 0.800 | 0.988 |
| | raw | NA | 1.000 | 1.000 |
| Tripathi et al. | analytical-VAE | 117789 | 0.845 | 0.820 |
| | stochastic-VAE | 13815 | 0.914 | 0.901 |
| | LDA | 132091 | 0.667 | 0.671 |
| | raw | NA | 0.571 | 0.632 |
| All Test Samples | analytical-VAE | 117789 | | |
| | stochastic-VAE | 13647 | | |
| | LDA | 169384 | | |

Table 2: Classification on held out data benchmarks. KNN classifiers were trained on data observed by the VAE, and evaluated on 1149 held out test samples. KNN with k=1 was used across all representations. Predictive log probability is also added to measure the generalizability of the VAEs across held out samples (smaller values are better).

Due to the relationship between squared Euclidean distance and proportionality given in Equation 12, we expect the proportionality metric to exhibit a log-linear relationship between the Euclidean distances obtained from the VAE embeddings. Indeed, this is apparent with the analytical VAE embeddings in Shalapour et al., and to a lesser extent in Tripathi et al., as shown in Figure S5.

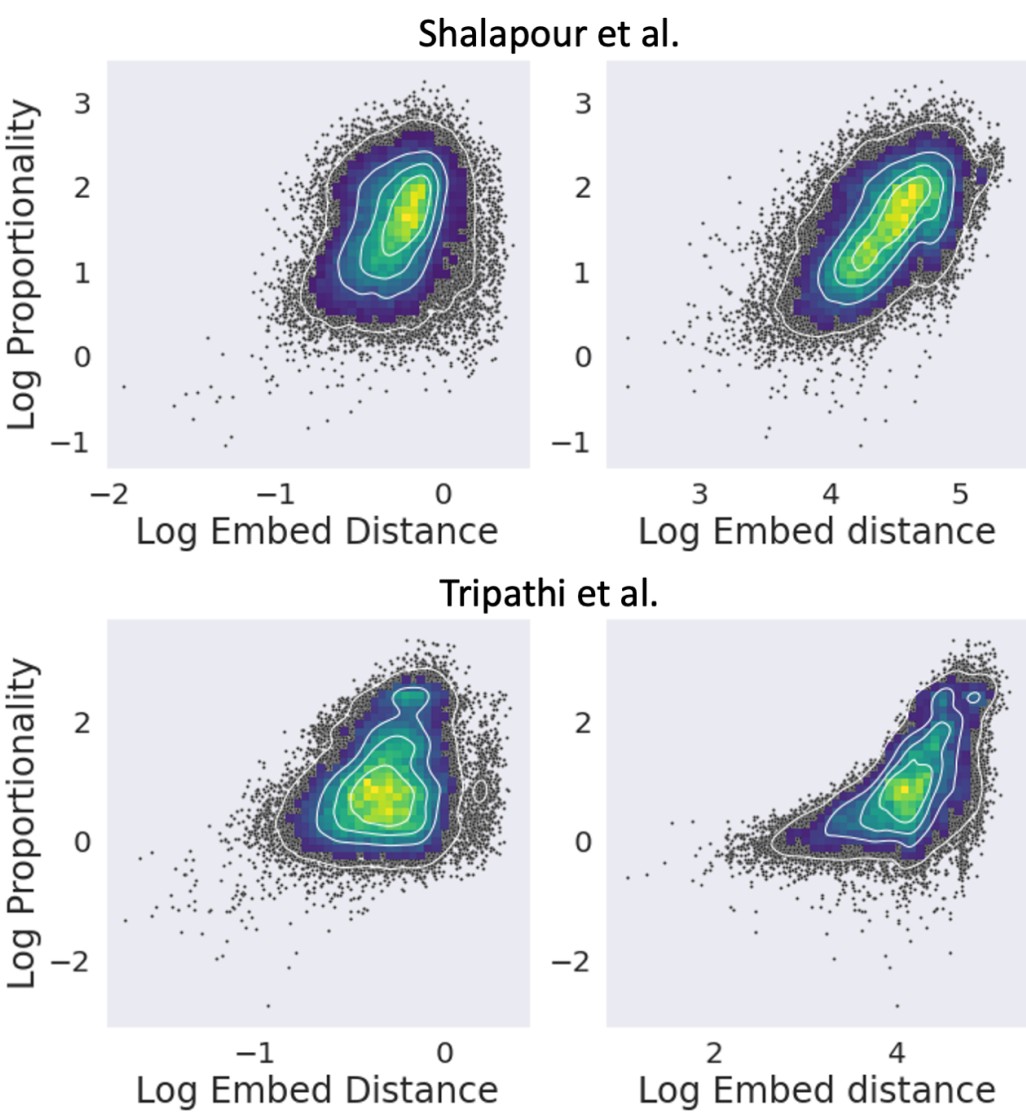

Figure S5: Comparison of Lovell's proportionality metric and the pairwise embedding distances for the stochastic VAE and the analytical VAE evaluated on the Shalapour et al. dataset and the Tripathi et al. dataset. The embedding distance is defined by the Euclidean distance between two rows within $\Psi W$. Since the proportionality metric is not defined for zeros, it is only evaluated for samples where both taxa are observed. Only the top 300 most abundant microbes are visualized here.

