# OpenReview forum: "Inferring Principal Components in the Simplex with Multinomial Variational Autoencoders"
_ICLR.cc/2021/Conference — Reject_

### Official Review · AnonReviewer1 · 2020-10-26
**Multinomial Variational Autoencoders can recover Principal Components**

**Rating:** 5
**Confidence:** 3

**Review:**

### Summary
The authors propose a framework to estimate high-dimensional covariance matrices using Multinomial variational autoencoders, showing application to biomedical data sets.
In particular, they use the probabilistic PCA framework, extending it to multinomial-distributed data. They show that similar work on PPCA are limited to Gaussian data, and non-easily applicable to bag-of-words. Analysis on real-world biomedical data sets are promising.

### Reasons for score
I fail to grasps the novelty with respect to the state of the art. The authors cite numerous of similar works, such as mixture modelling (LDA). The authors mention that such techniques rely on Dirichlet distributions, but that is not always the case (such as in [1] and [2]), where they show it can be approximated with a Logistic Normal. It seems such models can be used in the applications mentioned by the authors. However, since there are no comparison in the paper, I fail to understand the benefit of one with respect to the others.

### Pros
- ILR transform to deal with identifiability of softmax
- Analysis on synthetic and real data sets

### Cons
- Technical details can be improved. For example, it is not straightforward to understand what the encoder / decoder parameters refer to (is the type of activation functions at each layer? dimension of layers? number of layers?)
- There is no comparison with mixture models that infer covariance matrices (like [1] and [2]). It is difficult to assess what are the benefit of one with respect to the others.

### Minors
- "thethe", page 5

### References
[1] Srivastava, Akash, and Charles Sutton. "Autoencoding variational inference for topic models." arXiv preprint arXiv:1703.01488 (2017).
[2] Blei, David, and John Lafferty. "Correlated topic models." Advances in neural information processing systems 18 (2006): 147.

---

> ### Author Response · Authors · 2020-11-22
> **Multinomial VAEs outperforms LDA**
>
> Thank you for your comments.
>
> We have added a schematic figure in the supplemental materials to showcase the VAE neural network architecture.
>
> It is important to note that the methodology proposed here is quite different from what has been proposed in the LDA literature; we are proposing an analogue to principal components analysis to count-distributed data, whereas LDA is primarily a clustering technique.   As a result, these two methods are very different, yielding completely different interpretations.  That being said, we are able to benchmark their representations in the classification benchmark (see Table 1).
>
> The Multinomial PCA and Correlated topic models are quite different models.  If applied to documents, Multinomial PCA would infer word-word co-occurrences from the covariance matrix, whereas Correlated Topic models utilizes the covariance matrix to infer topic covariances.  It is possible to force a Correlated Topic model to fit the structure described in our paper, but it would require there to be an equal number of topics as words, and each topic only assigned to 1 word.  This is an edge case that cannot be handled with the current software packages.
>
> The innovation here is that we have provided a means to enable Multinomial PCA at scale, and have shown this model has stronger explanatory power than LDA in terms of held-out predictive log-likelihood and classification accuracy.

---

### Official Review · AnonReviewer2 · 2020-10-28
**A great interdisciplinary work**

**Rating:** 7
**Confidence:** 3

**Review:**

This paper is well written, presenting a great interdisciplinary work on covariance estimation. It relies on recent techniques such as connection between VAE and PPCA, ILR transformation, and presents an augmented VAE to obtain MAP estimates on multinomially distributed data. Such setting has direct application potential in many bioinformatics problems. The methodology is quite dense which reads look to me (though not in every detail).

The only chance it gets rejected is its relatively narrow audience group in ICLR.

1. the notation can be improved to fit community convention: say P(z|\theta, x), it's better to read P(z|x; \theta) since \theta are only parameters, not the target random variable.
2. Clarity can be improved on the part elaborating Algorithm 1, to clarify conclusive statements, eg justification on complexity, and put aside technical details. Readers may have confusions on how ILR transformation is used here: maybe Section 4.2 can hint on the exact use case and the part after Algorithm 1 can be better organized.
3. Experiments part should have more clear intro on machine learning abstracted versions of the problems, which better fit general audience in ICLR.
4. Some minor typos, eg Section 4.1 Equation (2) should be I_{d-1} instead of I_d.

Hope this paper gets attention and wide adoptions on related biological applications.

---

> ### Author Response · Authors · 2020-11-22
> **Multinomial VAEs have applications for compositional data**
>
> Thank you for your comments. We have addressed these comments in the manuscript and have provided more background information behind compositional data analysis to highlight how these concepts can be extended to beyond microbiome applications.

---

### Official Review · AnonReviewer4 · 2020-10-28
**New solution to an important problem. Needs a little additional work to demonstrate effectiveness.**

**Rating:** 6
**Confidence:** 3

**Review:**

The authors demonstrate a VAE model and estimation framework with which the PPCA subspace is recovered for data with multinomial observations. The authors specifically are interested in the high-d scenario and show that their analytical results out-perform other VAE methods.

## General comments
The paper offers a novel solution to the seemingly persistent problem of performing efficient inference of a non-congugate latent variable model for count data. The authors claim that existing solutions are not scalable. The paper is clearly written with a few minor typos. I have 2 complaints:
1) The benchmarking analysis reports only relative performance across other VAEs. There are no other methods even considered and it would be nice to have some sense of what a "good" result looks like in an absolute sense (eg. what's a good score on subspace distance for 200 or 1000 input dimension problem? What would you get if you didn't use the VAE approach?).
2) Could the authors comment more on the scalability problems they are talking about? There appear to be a number of scalable PCA techniques available where the full covariance matrix need not be formed and decomposed in memory. Would none of those approaches  be adaptable the their problem?

## minor note
The authors mention a few papers where MCMC was proposed as a means to obtain a posterior over the latent variables but these ventures were dismissed as suffering from the curse of dimensionality (or something to the effect of that argument). One outstanding oversight to this collection of papers is [Linderman et al.](https://papers.nips.cc/paper/5660-dependent-multinomial-models-made-easy-stick-breaking-with-the-polya-gamma-augmentation) wherein the stick-breaking construction of the multinomial is coupled with Polya-gamma augmentation to offer a highly efficient sampling procedure for precisely the model the authors are proposing. Could the authors comment in the rebuttal on how this approach

---

> ### Author Response · Authors · 2020-11-22
> **Additional baselines have been added to showcase effectiveness**
>
> Thank you for your comments.
>
> We have clarified how to interpret these metrics and have shown on fully observed count data that the ground truth covariance can be optimally estimated up to scale using VAEs.
>
> We did have a citation to the Polya-gamma augmentation approach proposed by Linderman (see reference 32).  The issue with this approach is that it is sensitive to the ordering of the labels, and consequently is not permutation invariant, a fundamental property that must be satisfied for compositional data.  Our approach addresses this through the rotation invariance of PCA.
>
> Regarding the comment on scalability challenges, we have included additional simulation benchmarking using the VAEs to initialize parameters using Hamiltonian Monte Carlo.  On the smallest dataset, it took over 24 hours to fit HMC, compared to the 20 minutes it took to fit the VAE models, and as shown in our benchmarks, the results are indeed comparable. Even on our smallest simulation datasets, fitting HMC without a strong initialization is challenging.
>
> Regarding the comment on scalable PCA approaches, yes there is likely future work in this direction, in addition to possible connections between online PCA and VAEs.  It wasn’t immediately obvious to us how to provide these extensions to fit these models in a compositional framework.

---

### Official Review · AnonReviewer3 · 2020-10-28
**weak experimental support of the paper's claim**

**Rating:** 4
**Confidence:** 3

**Review:**

This paper extends prior results, namely that VAEs are able to learn the principal components. The novelty is the extension to a new distribution: multinomial logistic-normal distribution. This is achieved by using the Isometric log-ratio (ILR) transform. While prior results were derived analytically, this paper provides (only) empirical evidence for the claim regarding the multinomial logistic-normal distribution.

Overall, I don’t find that the provided experiments provide convincing support of  the claim of the paper, namely that VAEs are able to learn the principle components for the multinomial logistic-normal distribution.

While the proposed approach yields better results than alternative approaches in Figure 1, it is not clear to me why the shown results indicate that the VAE was actually able to learn the principal components: based on my understanding of the used metrics (for instance,  axis-alignment of 0 indicates perfect results, and 1 is the worst): the proposed approach achieves 0.8 on 200 dimension and 0.95 on 1000 dimensions. When a deeper model is used in Figure 1, the metrics go down, which is good, but they still stay above 0.5, i.e., far away from 0. Based on my understanding of the used metrics, I am not sure why this would provide empirical support of  the claim in this paper that the principal components can be recovered.

In the definition of axis alignment in the appendix, it seems like the enumerator is missing a square, compared to the original definition in the referenced paper [14].

The authors also claim that Figure 2 shows that the learned embedding dimensions are orthogonal to each other. While I can believe this in the right plot, I am not convinced by the left plot, where a different scaling is used (as mentioned by the authors), and hence the diagonal does not seem so much larger than the off-diagonals to me.

The paper also proposes a batch-correction as additional improvement, but Table 1 shows that it makes results  actually worse.

Apart from that, Table 1 would also benefit from adding some state-of-the-art baseline models, e.g., as discussed in the related works section. Simply applying KNN to the raw counts seems like a very basic baseline.

Apart from that, the writing of the paper could be improved a lot.  I found it very confusing to come across references to Figure 5 etc,  and then not finding them in the paper ... eventually I realized that they were in a separate supplement. This could be made clearer in the paper. Moreover, there are also several typos in the paper (e.g., “thethe”), and some grammatically incorrect sentences. Also,  in the related works section, the authors refer to work by  “Luca et al” without a citation--does it  refer to Lucas (with an s) [4] ?

++++ updates after discussion period ++++

While the initial paper provided empirical evidence  for the claims of the paper, based on the reviews an appendix of about 9 pages was added in the revised version, and the focus now seems on providing theoretical support of the claims in the paper. Also several experimental results were added in the main paper in response to the reviews. I feel like all these (quite major) changes indicate that this paper is not mature enough for publication at this point. So I maintain my current review.

---

> ### Author Response · Authors · 2020-11-22
> **Experimental details have been updated showing our conclusions still hold**
>
> We have provided analytical insights to the Multinomial Logistic Normal distribution (see supplemental Appendix B) -- since the multinomial logistic normal distribution is globally log-convex, it must have a global optimum. As a result, augmenting PPCA with the multinomial distribution will not introduce additional local optima.   Furthermore, similar arguments provided by Lucas et al and Kunin et al also apply here; something that was not realized in earlier works due to the conditional non-conjugacy of the Multinomial Logistic Normal.
>
> Regarding the simulation results, the axis-alignment metric appears to be problematic in the context of count data for reasons that are currently unknown -- we have shown that even Multinomial PPCA fitted with Hamiltonian Monte Carlo cannot obtain axis-alignment metrics that approach zero.   Resolving this question is currently outside the scope of this paper and we will defer this problem for future work. However, we have shown strong agreement with the ground truth and estimated covariances when count data is fully observed (this is now shown in Figure 1).  Estimating principal components on sparse data is a challenging outstanding problem, which we have elaborated on in the text.
>
> Regarding the heatmap of the decoder inner product; we have extended discussions on this in addition to visualizing the distribution of diagonal elements (it turned out we accidentally included the wrong model, now the distinction between the embeddings learned from these two different methodologies is more apparent).
>
> Regarding the comment about the classification benchmarks, we added LDA to serve as another baseline for comparison.
>
> Dealing batch effects is a tricky problem that we have decided is outside the scope of this paper, so we have removed these benchmarks to be fleshed out more fully in future work.

---

### Decision · Program_Chairs · 2021-01-07
**Final Decision**

**Decision:**

Reject

**Comment:**

Authors extend the probabilistic PCA framework to multinomial-distributed data. Scalable estimation of principal components in the model is achieved using a multinomial variational autoencoder in combination with an isometric log-ratio (ILR) transform.
The reviewers did not agree on the degree of novelty of the paper to PC estimation.
The presentation of the paper can be improved.
The reviewers criticise that large changes have been made to the paper during the rebuttal phase.
Overall, the paper is borderline and due to the mentioned large changes I recommend a rejection (and re-review at a different venue).